# Effect of the COVID-19 Pandemic on Physical Activity and Sedentary Behavior in Older Adults

**DOI:** 10.3390/jcm11061568

**Published:** 2022-03-12

**Authors:** Elizabeth C. Lefferts, Joseph M. Saavedra, Bong Kil Song, Duck-chul Lee

**Affiliations:** Department of Kinesiology, Iowa State University, Ames, IA 50014, USA; eschro@iastate.edu (E.C.L.); joeysav@iastate.edu (J.M.S.); bksong@iastate.edu (B.K.S.)

**Keywords:** exercise, muscle-strengthening physical activity, coronavirus, United States, physical-activity guidelines

## Abstract

Whether the COVID-19 pandemic has long-lasting effects on physical activity (PA) and sedentary behavior in the vulnerable older adult population is uncertain. A total of 387 older adults (75 ± 6 years) completed a retrospective questionnaire on time spent sitting, walking, and performing aerobic and muscle-strengthening PA before, during the first three months, and one year into the COVID-19 pandemic. Whether the participants met the aerobic and muscle-strengthening PA guidelines was then determined. Of the 387 older adults, 376 (97%) were vaccinated. The participants completed 361 ± 426, 293 ± 400, and 454 ± 501 min/week of moderate-to-vigorous aerobic PA before, during the first three months, and one year into the pandemic, respectively. During the same time periods, the participants performed muscle-strengthening PA 87 ± 157, 68 ± 163, and 90 ± 176 min/week, walked 2.4 ± 1.7, 2.3 ± 1.7, and 2.6 ± 1.9 h/day, and sat 6.2 ± 2.9, 7.4 ± 3.1, and 6.1 ± 2.9 h/day, respectively. Aerobic PA, muscle-strengthening PA, and walking time decreased, whereas sitting time increased, during the first three months of the pandemic (*p* < 0.05), and then returned to pre-pandemic levels after one year (*p* < 0.05). The percentage of participants meeting both aerobic and muscle-strengthening PA guidelines decreased during the first three months of the pandemic (48.9% to 33.5%, *p* < 0.001), but returned to pre-pandemic levels one year later (*p* < 0.001). In conclusion, the COVID-19 pandemic significantly decreased PA and increased sitting time in older adults; however, both PA and sitting time returned to pre-pandemic levels after one year.

## 1. Introduction

On 11 March 2020, the World Health Organization (WHO) officially declared SARS-CoV-2 (COVID-19) a global pandemic. Shortly thereafter, several countries enacted mitigation strategies to reduce transmission such as physical distancing, stay-at-home orders, and the closing of non-essential businesses. The implementation of such strategies reduced the transmission of COVID-19 [1]; however, they also drastically altered day-to-day life, impacting normal activities such as physical activity.

The United States (US) Physical-Activity Guidelines [2], as well as those of the WHO [3], recommend that adults accumulate at least 150 min of moderate-intensity, 75 min of vigorous-intensity, or an equivalent combination of moderate- and vigorous-intensity aerobic physical activity per week, as well as at least two days per week of muscle-strengthening activity, to experience substantial health benefits. Meeting the physical-activity guidelines or performing regular physical activity extends additional health benefits specifically for COVID-19 as recent data suggest that these individuals have a lower risk of infection, hospitalization, intensive-care-unit admission, and mortality from COVID-19 [4,5,6]. The benefits of physical activity for COVID-19 may be related to how regular physical activity impacts multiple health domains that are related to COVID-19 risk, such as improving cardiovascular health [7], lowering systemic inflammation [8,9], improving immune health [10,11], and lowering obesity [12]. However, reduced physical-activity levels during the initial stages of the pandemic have been reported globally and across all age groups [13,14,15,16,17]. 

Older adults in particular are at a higher risk for COVID-19 [18,19] and are more susceptible to chronic diseases related to physical inactivity [20,21], making greater physical activity and lower sedentary time vital for their health. Recent evidence suggests that the COVID-19 pandemic has disproportionately impacted the physical activity of older adults compared to younger adults [16], with approximately 25–50% of older adults reporting a reduction in physical activity during the initial stages of the COVID-19 pandemic [22,23,24,25]. However, little is known about the changes in physical activity and sedentary behavior in older adults beyond the initial stages of the pandemic, especially after the COVID-19 vaccine became available. 

Starting in December 2020, vaccines against COVID-19 started to become available in the US and many of the restrictions had either eased or been completely retracted. After large initial decreases in physical activity at the pandemic onset in March 2020 in the US, visible increases in step counts were observed by June 2020 [14] and the partial recovery of physical activity was reported [26,27]. However, there are little data on physical activity and sedentary behavior, especially in the vulnerable older adult population, beyond January 2021 [27], thus making it difficult to assess whether the COVID-19 pandemic has long-lasting effects on physical activity and sedentary behavior. The aim of this study was to examine the changes in time spent sitting, walking, and performing aerobic and muscle-strengthening physical activity before the COVID-19 pandemic, during the first three months of the pandemic, and one year into the pandemic, when the vaccine was readily available and widely administered in the US to older adults, and several pandemic restrictions had been lifted. We hypothesized that physical activity would decrease and that sedentary time would increase during the first three months of the COVID-19 pandemic, but would return to pre-pandemic levels one year into the pandemic. 

## 2. Materials and Methods

### 2.1. Study Design, Population, and Recruitment

Data were collected as part of the Physical Activity and Aging Study (PAAS), an ongoing prospective cohort of men and women at least 65 years old conducted in Ames, Iowa, United States. PAAS participants are recruited from a college town and are primarily highly educated, white, retired, and living independently [28]. Since 2015, 755 older adults have completed at least one study visit. The currently enrolled participants in PAAS were recruited via e-mail and postal mail to participate in this one-time survey related to physical activity prior to and during the COVID-19 pandemic, in addition to COVID-19 history and vaccination status. Individuals were eligible for study participation if they were 65 years or older. Recruitment was completed in two phases. First, an e-mail invitation with a link to an online survey was sent via REDCap, an electronic data-capturing tool [29,30], to 659 active participants with valid e-mail addresses in June 2021, of which 297 participants completed the survey. Second, a paper survey was sent via postal mail to 302 individuals with valid mailing addresses who did not respond to the initial e-mail invitation in July 2021, of which 100 participants returned the survey. The overall response rate (e-mail and postal mail) was 60.2% (*n* = 397). Participants were excluded from the analyses for missing all physical-activity data (*n* = 2), changes in health status other than COVID-19 during the study period dramatically impacting physical activity such as a broken hip and initiation of cancer treatment (*n* = 4), and missing COVID-19 vaccination status (*n* = 4), resulting in a final sample of 387. All participants provided written informed consent and the study was approved by the Iowa State University Institutional Review Board.

### 2.2. Physical Activity

Physical activity was assessed via a questionnaire (Appendix A) that was developed based on the International Physical Activity Questionnaire (IPAQ). Participants were provided the questionnaire in June/July 2021 and asked to retrospectively recall their average weekly physical-activity levels from three different time points: three months before (December 2019–February 2020), during the first three months (March 2020–May 2020), and one year into the COVID-19 pandemic (April 2021–June 2021), when the COVID-19 vaccine was readily available and widely administered in US older adults, and several pandemic restrictions had been lifted (Figure 1). For each time point, the questionnaire asked how many times per week participants performed moderate, vigorous, and muscle-strengthening physical activity and the average duration each time, in minutes. The number of minutes per week spent completing moderate, vigorous, and muscle-strengthening physical activity was determined by multiplying the times per week by the duration in minutes. Following IPAQ guidelines, data were truncated at 21 h/week of moderate, vigorous, or muscle-strengthening physical activity [31]. In addition, participants were asked to report the total number of hours and minutes spent sitting and walking on both week days and weekend days. The average hours per day of sitting and walking were calculated as (hours on week days × 5 + hours on weekend days × 2]/7. Due to sparse data, we truncated the analysis at 9 h/day of walking (99% of distribution) [32].

Whether participants met the aerobic, resistance, or both physical-activity guidelines was also determined based on the reported physical activity [2]. Total moderate-to-vigorous physical activity (MVPA) was calculated as: (min of moderate physical activity) + 2 × (min of vigorous physical activity). Time spent walking was not included in the calculation of total MVPA following a previous study using the IPAQ on older adults [33]. Individuals who achieved ≥150 min/week of total MVPA were considered to meet the aerobic physical-activity guidelines [2]. Participants who performed muscle-strengthening physical activity ≥2 times per week were considered to meet the muscle-strengthening physical-activity guidelines [2]. Last, we determined the number of participants meeting both the aerobic and muscle-strengthening physical-activity guidelines.

### 2.3. COVID-19 Diagnosis and Vaccination Status

Previous COVID-19 diagnosis was assessed via questionnaire by asking “have you ever been diagnosed (tested positive) with COVID-19?”. If yes, participants were asked the month and year of diagnosis and whether they were hospitalized. Similarly, vaccination status was assessed via questionnaire by asking “have you received the COVID-19 vaccine?”. If yes, participants were asked when they first received the vaccine (i.e., first dose if a two-dose vaccine). For individuals who did not receive the vaccine, it was indicated whether it was due to personal choice, medical reasons, or other. 

### 2.4. Covariates

Participants were asked to recall their weight immediately before the onset of the COVID-19 pandemic and report their current weight at the time of the questionnaire (June 2021–July 2021). Body-mass index (BMI, kg/m^2^) was calculated as weight (kg) immediately before the COVID-19 pandemic/height (m^2^) from the most recent in-person visit (28 ± 11 months prior). The self-reported weight immediately before the COVID-19 pandemic was strongly correlated with the most recently measured weight within the cohort (r = 0.96), with a mean difference of 0.4 ± 4.5 kg. 

The presence of co-morbidities was assessed via self-reporting at the time of the survey by asking participants whether a physician had ever diagnosed them with hypertension, stroke, heart attack, diabetes, high cholesterol, chronic obstructive pulmonary disease, asthma, or cancer, which are common conditions associated with greater severity of illness from COVID-19. A summated score was then created from 0 (no co-morbidities) to 8 (all co-morbidities) and used in analyses.

### 2.5. Statistical Analysis

Descriptive means, standard deviations, and percentages are reported for the full sample. We used a linear mixed-effects model with repeated measures to assess the change in continuous variables (sitting, walking, and physical activity) across 3 time points (before the pandemic, the first three months of the pandemic, and one year into the pandemic). All models were adjusted for age, sex, BMI before the COVID-19 pandemic, and co-morbidities. The autoregressive covariance structure was selected for the model by determining the best fit according to the lower values for information criterion, after evaluation of different structures. We used a Cochran’s Q test to assess differences in the categorical variables of meeting the physical-activity guidelines across the 3 time points. When an overall significant difference was observed, McNemar’s test was used to test differences between paired time points (i.e., before the pandemic vs. first three months of the pandemic; first three months of the pandemic vs. one year into the pandemic; before the pandemic vs. one year into the pandemic).

For each individual physical-activity and sedentary-behavior outcome (sitting, walking, MVPA, etc.), only participants with complete data at all 3 time points were included in the analyses in order to maximally utilize the available data. This resulted in different sample sizes for each analysis by outcome. We further performed stratified analyses for sex (male, female), age (<75 years, ≥75 years), BMI (normal weight [<25 kg/m^2^], overweight/obese [≥25 kg/m^2^]), meeting both aerobic and muscle-strengthening physical-activity guidelines before the COVID-19 pandemic (meet, not meet), and co-morbidities (0–2, ≥2) on all continuous outcomes. We additionally performed sensitivity analyses adjusting for positive COVID-19 diagnosis and having received the COVID-19 vaccination. This adjustment was not part of our main model, as although these events may impact physical activity, they would not have equal effect at each of the 3 time points assessed (e.g., participants were vaccinated right before the third time point of one year into the pandemic, but not earlier for the first 2 time points). Statistical analyses were performed using SAS software version 9.4 (SAS Institute, Cary, NC, USA). All *p*-values are two-sided with an a priori α-level of 0.05 deemed significant.

## 3. Results

Descriptive characteristics of the sample are provided in Table 1. Among the 387 older adults, 376 (97.2%) had received a COVID-19 vaccine at the time of the survey, with 92% having started the vaccination process by April of 2021. Eleven participants had not received the vaccine due to personal choice (*n* = 8), medical reasons (*n* = 1), and other (*n* = 2). Twenty participants (5.2%) reported a previous diagnosis of COVID-19 with only two COVID-19 cases resulting in hospitalization. No significant change in body weight (76.6 ± 16.2 to 76.4 ± 16.3 kg, *p* = 0.42) was observed from before the COVID-19 pandemic to the present.

### 3.1. Physical Activity and Sitting Time

Figure 2 shows changes in physical activity and sitting levels over time. Before the COVID-19 pandemic, older adults reported sitting for 6.2 ± 2.9 h/day, walking 2.4 ± 1.7 h/day, and performing 212 ± 230, 78 ± 147, 361 ± 426 and 87 ± 157 min/week of moderate-intensity aerobic physical activity, vigorous-intensity aerobic physical activity, total MVPA, and muscle-strengthening physical activity, respectively. The hours per day spent sitting significantly increased during the first three months of the pandemic (7.4 ± 3.1 h/day) and returned to levels similar to before the pandemic one year later (6.1 ± 2.9 h/day) after adjusting for age, sex, BMI before the COVID-19 pandemic, and number of co-morbidities. A similar pattern was observed for muscle-strengthening physical activity, which decreased (68 ± 163 min/day) during the first three months of the pandemic and then returned to pre-pandemic levels one year into the pandemic (90 ± 167 min/day). In contrast, time spent walking significantly decreased during the first three months of the pandemic (2.3 ± 1.7 h/day) and then exceeded pre-pandemic values at one year into the pandemic (2.6 ± 1.9 h/day). Moderate-intensity, vigorous-intensity, and total MVPA followed similar patterns as walking, with significant reductions during the first three months of the pandemic (moderate: 184 ± 229 min/week; vigorous 56 ± 125 min/week; total MVPA 293 ± 400 min/week) but greater physical activity observed after one year into the pandemic (moderate: 261 ± 288 min/week; vigorous 102 ± 171 min/week; total MVPA 454 ± 501 min/week) compared to before the pandemic (*p* < 0.001). There were no changes in any of the reported results after additional adjustment for positive COVID-19 diagnosis and vaccination.

The stratified analyses (Table 2) show similar patterns between males and females, participants older than 75 and younger than 75, and participants with less than two or more than two co-morbidities (*p* > 0.05 for all interactions). An interaction effect was observed for sitting time (*p* = 0.04) between normal weight and overweight/obese individuals, suggesting a greater increase in sitting time for overweight/obese individuals during the first three months of the pandemic. In individuals who did and did not meet both physical-activity guidelines at baseline, no interaction effects were observed for sitting or walking (*p* > 0.05); however, significant interaction effects were observed for total MVPA (*p* < 0.001) and muscle-strengthening physical activity (*p* = 0.03). These interactions suggest that individuals who met the physical-activity guidelines had larger reductions in both aerobic and muscle-strengthening physical activity during the first three months of the pandemic, whereas no significant change was observed in those who did not meet the physical-activity guidelines.

### 3.2. Physical-Activity Guidelines

Before the COVID-19 pandemic, 67.9%, 61.9%, and 48.9% of older adults met the aerobic, muscle-strengthening, and both physical-activity guidelines, respectively (Figure 3). The percentage of older adults meeting aerobic, muscle-strengthening, and both physical-activity guidelines significantly decreased to 54.8% (−13.1%), 46.1% (−15.8%), and 33.5% (−15.4%), respectively, during the first three months of the pandemic (all *p* < 0.001). However, one year into the pandemic, the number of older adults meeting the physical-activity guidelines returned to levels similar to before the pandemic (71.1% aerobic, 61.3% muscle-strengthening, 50.5% both).

## 4. Discussion

We observed significant reductions in time spent walking and performing aerobic and muscle-strengthening physical activity, with a concomitant increase in time spent sitting, during the first three months of the COVID-19 pandemic in our older adults. However, one year into the pandemic, when the COVID-19 vaccine was readily available and administered for most participants and restrictions lifted, all physical activity and sitting time had either recovered to levels similar to or exceeded those reported before the COVID-19 pandemic. It appears that the pandemic adversely altered the physical activity and sedentary behavior in our older adults; however, our data optimistically suggest a return to regular physical-activity levels and sedentary behavior among older adults. 

### 4.1. Physical Activity and Sitting Time

Our findings of decreases in physical activity and increases in sitting time during the first three months of the COVID-19 pandemic in older adults confirm the findings of previous studies on adults, suggesting the COVID-19 pandemic was accompanied by reductions in physical activity [13,15,16,34,35]. Wilke et al. [16] studied physical activity during the COVID-19-pandemic restrictions across different adult age groups and countries, finding that all age groups decreased both moderate and vigorous physical activity by at least 35%, and the reduction in physical activity was observed in countries impacted by COVID-19 all over the world, such as Brazil, France, Italy, Singapore, and South Africa. Previous studies on adults have also suggested physical-activity levels were starting to increase within six months of the initial decline following the onset of the COVID-19 pandemic [14,26,34]. Joseph et al. [34] previously examined time spent walking and performing moderate, vigorous, and total physical activity in 589 adults 50 years or older from the US. They reported that by June/July of 2020, moderate-intensity physical-activity levels had returned to pre-pandemic levels, whereas time spent in vigorous-intensity physical activity had increased but not yet returned to pre-pandemic levels. Thus, in June/July of 2020, walking, vigorous, and total physical activity were still significantly lower than pre-pandemic levels. To our knowledge, we are the first to report physical-activity levels one year after the onset of the COVID-19 pandemic, particularly in older adults in which physical activity is a critical lifestyle behavior for COVID-19 and health. Our study expands upon these previous findings by Joseph et al. [34], suggesting that at one year into the COVID-19 pandemic, after the vaccine was available and administered to most older adults in the US and restrictions had lifted, physical activity at all intensities and sitting time had returned to or exceeded pre-pandemic levels. 

The observation of drastic reductions in physical activity during the first three months of the COVID-19 pandemic is alarming from a public health perspective; however, the recovery of these physical-activity levels within one year is encouraging. The potential cause of the increase in physical activity observed from the onset of the COVID-19 pandemic to one year later in these older adults is likely multi-factorial. By April of 2021, federal, state, and local governments removed many societal restrictions, including mask mandates. Individuals were more informed about how COVID-19 is transmitted and about different mitigation strategies they could employ for safety. Most importantly, widespread vaccine availability may have also played a critical role in increasing the comfort of older adults to return to their normal physical-activity routines. We cannot determine when exactly physical-activity levels returned to or exceeded the levels before the pandemic; however, the recovered physical-activity levels one year after the start of the pandemic is hopeful for the long-term impact on the health of older adults. Previous concern was raised that the pandemic “*would accelerate longitudinal declines in physical activity and that the majority of middle-aged and older adults will never again achieve their pre-pandemic physical-activity levels*” [34] based on a systematic review suggesting that physical-activity levels generally decline over time [36]. Our data provide an optimistic outlook that the reduction in physical activity and increase in sitting time during the COVID-19 pandemic is not long-lasting and is returning to baseline levels in older adults, likely due to multiple reasons, including policy changes and vaccination. 

It has also been hypothesized that the COVID-19 pandemic may create a secondary issue, recently termed ‘covibesity’ [37], suggesting that the reductions in physical activity, increases in stress, rise in alcohol consumption, and other negative lifestyle choices accompanying the COVID-19 pandemic will elicit rapid weight gain. Indeed, several studies suggest more rapid weight gain during the COVID-19 pandemic in adults [38,39,40], with one study indicating adults at least 18 years of age in the US increased their body weight 1% (0.6 kg) on average over three months, with almost 20% of individuals gaining more than 2 kg [40]. However, we did not observe any changes in self-reported weight in our older adult population from immediately before the pandemic to the present. Although we cannot determine whether appreciable weight changes occurred between these two time points, our data could suggest that either (1) the COVID-19 pandemic had minimal influence on weight in older adults or (2) similar to the pattern observed in physical-activity behavior, weight management recovered one year into the pandemic. Overall, our results suggest that the ‘covibesity’ phenomenon is less apparent in an otherwise healthy, older adult population, though we acknowledge that we do not have diet data in this study and future research is necessary to fully assess the implications of the COVID-19 pandemic related changes in physical activity and measured body weight over the long term. 

In our stratified analyses, we did not observe any differences in the changes in physical activity by either sex, age, or number of chronic conditions in older adults. However, when comparing normal weight and overweight/obese individuals, overweight/obese individuals had a larger increase in sitting time at the onset of the COVID-19 pandemic. Greater sedentary time is associated with higher all-cause and cardiovascular-disease morbidity and mortality [41,42]. Although sedentary time returned to baseline levels by one year after the pandemic onset, it may be of future interest to investigate how changes in sedentary time during the COVID-19 pandemic influenced all-cause and cardiovascular-disease morbidity and mortality. 

### 4.2. Physical-Activity Guidelines

The changes in physical activity were accompanied by fluctuations in the proportion of participants meeting the physical-activity guidelines. We observed a 13–16% reduction in older adults meeting the aerobic, muscle-strengthening, or both physical-activity guidelines at the onset of the COVID-19 pandemic (Figure 3). This magnitude of change is similar to the 18% drop in meeting the aerobic physical-activity guidelines that was reported in 13,503 adults aged 18 and older worldwide [16], as well as the 17% drop in approximately 9000 older adults 60 years or older in Brazil [35] from before to the early stages of the pandemic. However, one year after the pandemic, meeting the physical-activity guidelines recovered to levels similar to before the pandemic in our study in older adults. Wilke et al. [16] suggest that individuals who were more active before the COVID-19 pandemic saw the greatest reductions in physical activity during the pandemic. In our stratified analysis, we observed similar changes in sitting and walking in our older adults who did and did not meet both the physical-activity guidelines before the pandemic; however, we observed larger decreases in higher-intensity physical activity in older adults who met the physical-activity guidelines before the COVID-19 pandemic. This may be related to the restrictions imposed at the onset of the COVID-19 pandemic. For example, many fitness centers and group exercise classes were shut down or cancelled, which may have been one of the primary modes for older adults to complete their higher-intensity and muscle-strengthening physical activity.

### 4.3. Strengths and Limitations

The strengths of the present study include the long time frame over which physical activity was reported in comparison to most other studies that only reported physical activity during the early stages of the pandemic. However, our sample is not representative of the US older adult population and the results are not generalizable to the entire demographic of US older adults given the lack of diversity in this cohort, as participants are primarily highly educated, white, and living independently. However, the scope of the pandemic and the timing of COVID-19 mitigation strategies were similar to other regions in the US. 

Our methodology also presents limitations as it relied on self-reporting from over one year prior, leaving the data susceptible to response bias, over-reporting of physical-activity levels [43], and recall bias [44]. However, we found a significant, although not strong, correlation between self-reported MVPA at the most recent in-person visit and three months before the pandemic in the current survey (r = 0.37, *p* < 0.001). Further, among the 236 participants who met the aerobic physical-activity guidelines at the most recent in-person visit, 176 (75%) also reported meeting the aerobic physical-activity guidelines three months before the pandemic in the current survey. Although it is probable some recall bias exists, it appears older adults may have consistent routines, making the estimation of physical activity over one year prior by self-reporting easier and sufficient. Nevertheless, the main purpose of the current study was to examine changes in physical activity and sedentary behavior over time using the same physical-activity questions. Thus, the magnitude of systematic measurement error would theoretically be similar across the time points and minimally influence the findings. Second, the three months before the pandemic coincided with the winter season in the Midwest US, whereas the first three months of the pandemic and one year into the pandemic were during the spring season, which could cause potential seasonal effects on physical activity. However, this limitation may have minimal effects given that reductions in physical activity were observed during the first three months of the pandemic despite moving from the winter to spring season, and both the time points of the first three months of pandemic and one year into the pandemic are from similar seasons, making them more comparable. Additionally, the survey was not actually conducted in different seasons, but participants were instead asked about their physical activity retrospectively at different time points of the pandemic. In this study, we specifically wanted to examine the immediate and long-term impact of the pandemic on physical activity and sedentary behavior by asking short, three-month time periods immediately before and after the pandemic and one year later when the COVID-19 vaccine was readily available and administered to most older adults. Third, since only 11 (3%) older adults were not vaccinated in this study, we were not able to directly compare the changes in physical activity and sitting time between vaccinated vs. unvaccinated older adults. This high vaccination rate is not surprising, given ~92% of older adults in Iowa have received the COVID vaccine [45]. Fourth, the study is observational, meaning causal links between events and changes in physical activity cannot be established. 

## 5. Conclusions

In older adults, physical activity decreased and sitting time increased significantly at the onset of the COVID-19 pandemic. However, one year later, after the COVID-19 vaccine was readily available and administered in most older adults in this study and several restrictions had been lifted, physical activity and sitting time returned to levels similar to or exceeding those before the pandemic. The recovery of physical activity is an optimistic outcome that may influence the long-term health outcomes stemming from a drastic, albeit temporary, reduction in physical activity and a concomitant increase in sitting time among older adults. However, future research assessing the impact of the changes in physical activity and sedentary behavior during the COVID-19 pandemic on long-term health outcomes is warranted. In hopes of alleviating potential negative effects of reduced physical activity, it is important to continue to encourage and support interventions among older adults to maintain and increase physical activity in this vulnerable population with the greatest need for regular physical activity for health. 

## Figures and Tables

**Figure 1 jcm-11-01568-f001:**
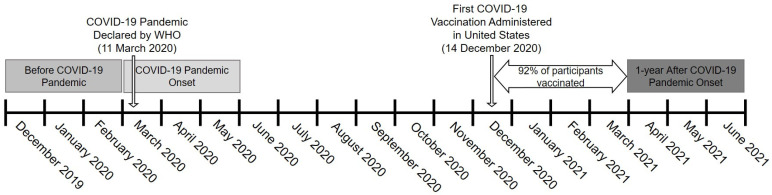
Study design depicting three different data-collection time points for physical activity and sedentary behavior. WHO, World Health Organization.

**Figure 2 jcm-11-01568-f002:**
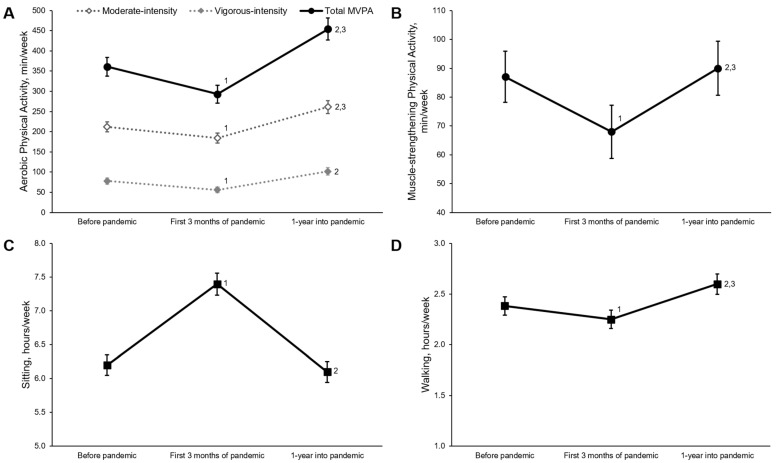
Changes in physical activity and sitting time during the COVID-19 pandemic. Data presented as mean and standard error for moderate-intensity, vigorous-intensity, and total moderate-to-vigorous intensity aerobic physical activity (**A**), muscle-strengthening physical activity (**B**), sitting (**C**), and walking (**D**). All analyses adjusted for age, sex, body-mass index before the COVID-19 pandemic, and number of co-morbidities. ^1^ Significant difference between before the COVID-19 pandemic and first three months of the COVID-19 pandemic, *p* < 0.05; ^2^ Significant difference between first three months of the COVID-19 pandemic and one year into the COVID-19 pandemic, *p* < 0.05; ^3^ Significant difference between before the COVID-19 pandemic and one year into the COVID-19 pandemic, *p* < 0.05. Sample sizes are 344, 355, and 336 for moderate-intensity, vigorous-intensity and total MVPA, respectively; 315 for muscle-strengthening physical activity; 360 for sitting time and 353 for walking.

**Figure 3 jcm-11-01568-f003:**
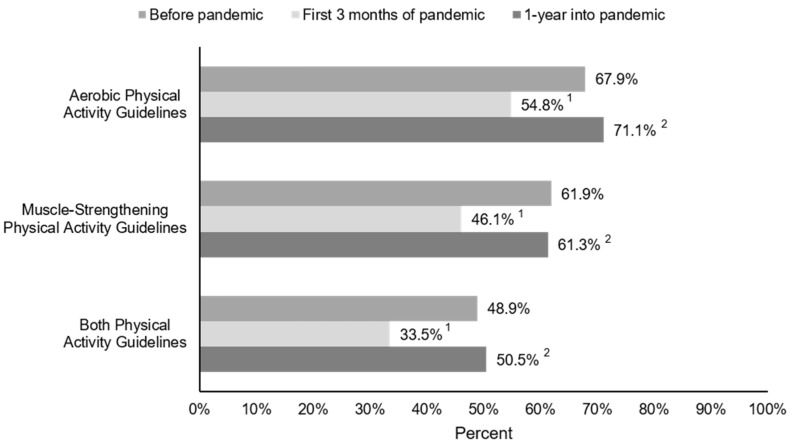
Changes in the percentage of older adults meeting physical-activity guidelines during the COVID-19 pandemic. ^1^ Significant difference between before the COVID-19 pandemic and first three months of the COVID-19 pandemic, *p* < 0.001; ^2^ Significant difference between first three months of the COVID-19 pandemic and one year into COVID-19 pandemic, *p* < 0.001. There were no significant differences in meeting the aerobic, muscle-strengthening, or both physical-activity guidelines before the pandemic and one year into the pandemic. Sample sizes are 336, 367, and 331 for aerobic, muscle-strengthening and both physical-activity guidelines, respectively.

**Table 1 jcm-11-01568-t001:** Demographic and clinical characteristics.

	*n* = 387
Age, years	75 ± 6
Female, *n* (%)	223 (57.6)
Height, cm	167.1 ± 10.4
BMI before COVID-19 pandemic, kg/m^2^	27.3 ± 5.0
Positive COVID-19 diagnosis, *n* (%)	20 (5.2)
Vaccinated against COVID-19, *n* %	376 (97.2)
Co-morbidities, number ^1^	1.4 ± 1.2
Hypertension, *n* (%)	135 (36.9)
Stroke, *n* (%)	12 (3.4)
Myocardial infarction, *n* (%)	15 (4.3)
Diabetes, *n* (%)	32 (9.2)
High cholesterol, *n* (%)	200 (54.1)
Chronic obstructive pulmonary disease, *n* (%)	11 (3.2)
Asthma, *n* (%)	33 (9.5)
Cancer, *n* (%)	89 (25.6)

Values are mean ± standard deviation or *n* (%). BMI, body-mass index. ^1^ Sum of co-morbidities present.

**Table 2 jcm-11-01568-t002:** Changes in physical activity and sitting time during the COVID-19 pandemic stratified by sex, age, obesity, co-morbidities, and meeting physical-activity guidelines before the pandemic.

			*n*	Before Pandemic (December 2019– February 2020)	First Three Months of Pandemic (March 2020– May 2020)	One Year into Pandemic (April 2021– June 2021)	Within-Group Time Effect *p*-Value ^a^	Interaction *p*-Value ^a^
Sex	Sitting, h/day	Male	157	6.4 ± 3.0	7.5 ± 3.1	6.3 ± 3.0	<0.001 ^1,2^	0.87
		Female	203	6.1 ± 2.8	7.2 ± 3.1	6.0 ± 2.8	<0.001 ^1,2^	
	Walking, h/day	Male	156	2.3 ± 1.7	2.2 ± 1.6	2.4 ± 1.7	0.001 ^1,2^	0.37
		Female	197	2.5 ± 1.7	2.3 ± 1.8	2.8 ± 2.0	<0.001 ^2,3^	
	Total MVPA, min/wk	Male	149	408 ± 431	354 ± 435	501 ± 509	<0.001 ^1,2,3^	0.51
		Female	187	323 ± 418	244 ± 364	407 ± 494	<0.001 ^1,2,3^	
	Muscle-strengthening PA, min/wk	Male	147	79 ± 147	65 ± 149	82 ± 151	<0.001 ^1,2^	0.61
		Female	168	94 ± 166	70 ± 176	96 ± 179	0.02 ^2^	
Age	Sitting, h/day	<75 years	215	6.2 ± 2.8	7.3 ± 3.0	6.0 ± 2.7	<0.001 ^1,2^	0.47
		≥75 years	145	6.3 ± 3.0	7.5 ± 3.3	6.3 ± 3.1	<0.001 ^1,2^	
	Walking, h/day	<75 years	215	2.3 ± 1.7	2.2 ± 1.8	2.6 ± 1.8	<0.001 ^2,3^	0.76
		≥75 years	138	2.5 ± 1.8	2.3 ± 1.7	2.6 ± 1.9	<0.001 ^1,2^	
	Total MVPA, min/wk	<75 years	206	238 ± 461	301 ± 410	468 ± 513	<0.001 ^1,2,3^	0.58
		≥75 years	130	328 ± 362	279 ± 384	432 ± 434	<0.001 ^1,2,3^	
	Muscle-strengthening PA, min/wk	<75 years	191	81 ± 118	55 ± 86	85 ± 138	<0.001 ^1,2^	0.11
		≥75 years	124	96 ± 204	88 ± 127	97 ± 204	0.72	
Obesity	Sitting, h/day	Normal weight	125	6.2 ± 3.1	7.1 ± 3.2	6.1 ± 3.1	<0.001 ^1,2^	0.04
		Overweight/Obese	235	6.2 ± 2.8	7.5 ± 3.1	6.1 ± 2.8	<0.001 ^1,2^	
	Walking, h/day	Normal weight	120	2.3 ± 1.7	2.1 ± 1.6	2.4 ± 1.8	0.001 ^2^	0.31
		Overweight/Obese	233	2.5 ± 1.7	2.3 ± 1.8	2.7 ± 1.9	<0.001 ^1,2,3^	
	Total MVPA, min/wk	Normal weight	114	449 ± 541	349 ± 489	515 ± 590	<0.001 ^1,2^	0.26
		Overweight/Obese	222	315 ± 344	264 ± 343	423 ± 448	<0.001 ^1,2,3^	
	Muscle-strengthening PA, min/wk	Normal weight	108	101 ± 142	66 ± 92	94 ± 151	<0.001 ^1,2^	0.14
		Overweight/Obese	207	79 ± 165	68 ± 191	87 ± 175	0.049 ^2^	
Co-morbidities	Sitting, h/day	<2	205	6.1 ± 2.9	7.1 ± 3.1	6.0 ± 2.9	<0.001 ^1,2^	0.09
		≥2	155	6.3 ± 2.9	7.6 ± 3.2	6.2 ± 2.9	<0.001 ^1,2^	
	Walking, h/day	<2	198	2.5 ± 1.8	2.4 ± 1.8	2.8 ± 2.0	<0.001 ^2,3^	0.38
		≥2	155	2.3 ± 1.6	2.1 ± 1.6	2.4 ± 1.6	0.001 ^1,2^	
	Total MVPA, min/wk	<2	185	388 ± 448	319 ± 434	482 ± 500	<0.001 ^1,2,3^	0.73
		≥2	151	327 ± 396	260 ± 394	433 ± 503	<0.001 ^1,2,3^	
	Muscle-strengthening PA, min/wk	<2	176	89 ± 161	69 ± 174	89 ± 175	0.06	0.80
		≥2	139	83 ± 153	65 ± 149	90 ± 156	<0.001 ^1,2^	
Meeting Both Aerobic and Muscle-strenghthening Physical-Activity Guidelines	Sitting, /day	Does Not Meet	167	6.4 ± 3.1	7.4 ± 3.3	6.2 ± 3.1	<0.001 ^1,2^	0.38
		Meet	158	6.0 ± 2.7	7.3 ± 3.1	6.0 ± 2.7	<0.001 ^1,2^	
	Walking, /day	Does Not Meet	162	2.3 ± 1.6	2.1 ± 1.6	2.5 ± 1.8	<0.001 ^1,2,3^	0.50
		Meet	158	2.5 ± 1.8	2.4 ± 1.8	2.7 ± 1.9	0.001 ^2^	
	Total MVPA, min/wk	Does Not Meet	170	208 ± 312	188 ± 295	317 ± 405	<0.001 ^2,3^	<0.001
		Meet	162	526 ± 470	407 ± 465	606 ± 552	<0.001 ^1,2,3^	
	Muscle-strengthening PA, min/wk	Does Not Meet	141	25 ± 45	27 ± 50	45 ± 91	0.003 ^2,3^	0.03
		Meet	156	126 ± 160	75 ± 188	117 ± 175	0.006 ^1,2^	

Values are mean ± SD. MVPA, moderate- and vigorous-intensity physical activity; PA, physical activity. ^a^ All analyses adjusted for age (not in age-stratified analyses), sex (not in sex-stratified analyses), BMI before COVID-19 pandemic (not in obesity-stratified analyses), and number of co-morbidities (not in co-morbidity-stratified analyses). ^1^ Significant difference between before the COVID-19 pandemic and first three months of COVID-19 pandemic, *p* < 0.05. ^2^ Significant differences between first three months of COVID-19 pandemic and one year into COVID-19 pandemic, *p* < 0.05. ^3^ Significant differences between before the COVID-19 pandemic and one year into COVID-19 pandemic, *p* < 0.05.

## Data Availability

The data presented in this study are available upon reasonable request from the corresponding author.

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
