# Peer review of "Effect of the COVID-19 Pandemic on Physical Activity and Sedentary Behavior in Older Adults"

_jcm, 2022, doi:10.3390/jcm11061568_

Round 1
Reviewer 1 Report
in my opinion is better to improve the highlight about the limitation of study> the first interview bout IPAQ was made remembering the past. this is presented as limitation but a strong declaration is better.
the authors can add some sentences in discussion about the comparison with different age samples (i.e. adolescent, children) or other countries people.
Author Response
We wish to thank the reviewer 1 for the time and attention given to our work and appreciate the thoughtful comments and suggestions. We have revised our work accordingly and responded to the reviewer comments below in a point-by-point fashion. Changes to the manuscript stemming from specific comments appear in red text within this document and track changes within the revised manuscript.
Reviewer 1 - Comments and Suggestions for Authors
Comment 1: “in my opinion is better to improve the highlight about the limitation of study> the first interview bout IPAQ was made remembering the past. this is presented as limitation but a strong declaration is better.”
Author Response: We have attempted to further clarify this point within our methodology by including the time frame when the questionnaire was administered and clarity that all 3 time points occurred on the same questionnaire: (Line 97) “Participants were provided the questionnaire in June/July 2021 and asked to retrospectively recall their average weekly physical activity levels from three different time points: 3 months before (December 2019 – February 2020), during the first 3-months (March 2020 – May 2020), and 1-year into the COVID-19 pandemic (April 2021 – June 2021)…”
Comment 2: “the authors can add some sentences in discussion about the comparison with different age samples (i.e. adolescent, children) or other countries people.”
Author Response: We have added the following sentence to our discussion in regard to physical activity across the adult age range and different countries: (Line 274) Wilke et al [16] studied physical activity during COVID-19 pandemic restrictions across different age groups and countries, finding all age groups decreased both moderate and vigorous physical activity by at least 35%, and the reduction in physical activity was observed in countries all over the world impacted by COVID-19, such as Brazil, France, Italy, Singapore, and South Africa.
Reviewer 2 Report
this study examines the effects of the covid epidemic confinement on physical activity in older adults. The topic is interesting and the study provides important, albeit partial, information (special population).
My main criticism concerns the delay between the collection of information and the confinement phase. The date of submission of the questionnaires is not clearly given but it is assumed that all the information was collected 1 year after the confinement. Therefore, some questions concern physical activity carried out more than one year ago, which poses a serious problem of reliability. It is necessary to clarify this point by specifying in the methodology the dates of the questionnaires and to reinforce the part of the discussion on the limits due to the delay of the questionnaires.
It would be interesting to know what restrictions were placed on unvaccinated individuals at the time of data collection. These restrictions are not well known to non-US readers. Also, is the vaccination rate in your sample within the norm for the US population?
Why did you adjust the models for age, gender, and BMI when you want to analyze change over time?
Author Response
We wish to thank the reviewer 2 for the time and attention given to our work and appreciate the thoughtful comments and suggestions. We have revised our work accordingly and responded to the reviewer comments below in a point-by-point fashion. Changes to the manuscript stemming from specific comments appear in red text within this document and track changes within the revised manuscript.
Reviewer 2 - Comments and Suggestions for Authors
Comment 1: “this study examines the effects of the covid epidemic confinement on physical activity in older adults. The topic is interesting and the study provides important, albeit partial, information (special population). My main criticism concerns the delay between the collection of information and the confinement phase. The date of submission of the questionnaires is not clearly given but it is assumed that all the information was collected 1 year after the confinement. Therefore, some questions concern physical activity carried out more than one year ago, which poses a serious problem of reliability. It is necessary to clarify this point by specifying in the methodology the dates of the questionnaires and to reinforce the part of the discussion on the limits due to the delay of the questionnaires.”
Author Response: We have better clarified in our methodology about the collection of physical activity that the questionnaires were provided to participants in June/July of 2021: (Line 97) “Participants were provided the questionnaire in June/July 2021 and asked to retrospectively recall their average weekly physical activity levels from three different time points: 3 months before (December 2019 – February 2020), during the first 3-months (March 2020 – May 2020), and 1-year into the COVID-19 pandemic (April 2021 – June 2021)…”
We have further added to our limitations section by overtly stating the time frame for the methodology in two locations: (Line 368) “relied on self-report over one year prior” and (Line 377) “making the estimation of physical activity over one year prior by self-report easier and sufficient”.
Comment 2: “It would be interesting to know what restrictions were placed on unvaccinated individuals at the time of data collection. These restrictions are not well known to non-US readers. Also, is the vaccination rate in your sample within the norm for the US population?”
Author Response: We thank the reviewer for the good comment. In regard to restrictions, at the time of the questionnaire, restrictions were not imposed based on vaccination status within the United States. Additionally, in Iowa, most COVID restrictions such as closed businesses, state imposed mask regulations, etc. had been removed by the time of the questionnaire, as stated throughout the manuscript as “several restrictions had been lifted” and very briefly mentioned on lines 298-301 (By April of 2021, federal, state, and local governments removed many societal restrictions, including mask mandates. Individuals were more informed about how COVID-19 is transmitted and different mitigation strategies they could employ for safety.). Because we did not look at the data by vaccination status and restrictions were not implemented based on vaccination status, we chose not to include several details about the different restrictions because they were regulated at a local level, suggesting the restrictions could be different for different participants within the study.
The vaccination rate within our sample, however, is representative of the vaccination rate for older adults (65+ years) within the state of Iowa (~92%) and the United States (~95%). We have added a sentence in regard to the vaccination rate to the manuscript: (Line 396) “This high vaccination rate is not surprising, given ~92% of older adults in Iowa have received the COVID vaccine [45].”
Comment 3: “Why did you adjust the models for age, gender, and BMI when you want to analyze change over time?”
Author Response: Age, sex, BMI, and comorbidities impact physical activity levels and sedentary time in the general population, thus are related to our primary outcomes. Although analyzing change over time, we needed to ensure these variables were not driving the effects of the analysis, thus they were included as covariates. Our stratified analyses suggest that age, sex, and comorbidities did not differentially effect the change in physical activity and sedentary time; however, overweight/obese individuals did have greater changes in sitting time compared to normal weight individuals.